# Heterogeneous Graph Learning for Visual Commonsense Reasoning

**Weijiang Yu**[1], **Jingwen Zhou**[1], **Weihao Yu**[1], **Xiaodan Liang**[2],[*] **Nong Xiao**[1]

[1]School of Data and Computer Science, Sun Yat-sen University
[2]School of Intelligent Systems Engineering, Sun Yat-sen University
weijiangyu8@gmail.com, zhoujw57@mail2.sysu.edu.cn, weihaoyu6@gmail.com,
xdliang328@gmail.com, xiaon6@sysu.edu.cn

## Abstract

Visual commonsense reasoning task aims at leading the research field into solving cognition-level reasoning with the ability of predicting correct answers and meanwhile providing convincing reasoning paths, resulting in three sub-tasks i.e., Q→A, QA→R and Q→AR. It poses great challenges over the proper semantic alignment between vision and linguistic domains and knowledge reasoning to generate persuasive reasoning paths. Existing works either resort to a powerful end-to-end network that cannot produce interpretable reasoning paths or solely explore intra-relationship of visual objects (homogeneous graph) while ignoring the cross-domain semantic alignment among visual concepts and linguistic words. In this paper, we propose a new Heterogeneous Graph Learning (HGL) framework for seamlessly integrating the intra-graph and inter-graph reasoning in order to bridge vision and language domain. Our HGL consists of a primal vision-to-answer heterogeneous graph (VAHG) module and a dual question-to-answer heterogeneous graph (QAHG) module to interactively refine reasoning paths for semantic agreement. Moreover, our HGL integrates a contextual voting module to exploit long-range visual context for better global reasoning. Experiments on the large-scale Visual Commonsense Reasoning benchmark demonstrate the superior performance of our proposed modules on three tasks (improving 5% accuracy on Q→A, 3.5% on QA→R, 5.8% on Q→AR)[2].

## 1 Introduction

Visual and language tasks have attracted more and more researches, which contains visual question answering (VQA) [29, 35, 36], visual dialogue [15, 10], visual question generation (VQG) [20, 30], visual grounding [17, 11, 41] and visual-language navigation [39, 21]. These tasks can roughly be divided into two types. One type is to explore a powerful end-to-end network. Devlin et al. have introduced a powerful end-to-end network named BERT [12] for learning more discriminative representation of languages. Anderson et al [2]. utilized the attention mechanisms [42] and presented a bottom-up top-down end-to-end architecture. While these works resorted to a powerful end-to-end network that cannot produce interpretable reasoning paths. The other type is to explore intra-relationship of visual objects (homogeneous graph). Norcliffe-Brown et al. [32] presented a spatial graph and a semantic graph to model object location and semantic relationships. Tang et al. [37] modeled the intra-relationship of visual objects by composing dynamic tree structures that place the visual objects into a visual context. However, all of them solely consider intra-relationship limited to homogeneous graphs, which is not enough for visual commonsense reasoning (VCR) due

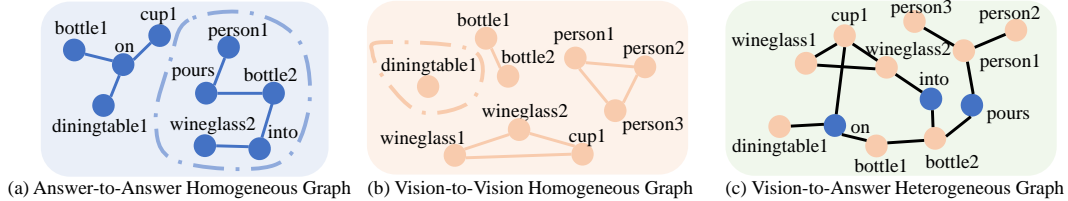

(a) Answer-to-Answer Homogeneous Graph    (b) Vision-to-Vision Homogeneous Graph    (c) Vision-to-Answer Heterogeneous Graph

Figure 1: (a) Answer-to-answer homogeneous graph is to model intra-relationship of each word in all answers of linguistics; (b) Vision-to-vision homogeneous graph is to mine intra-relationship of each object from images; (c) Vision-to-Answer heterogeneous graph is to excavate inter-relationship between object and answer. The dotted portion in (a)&(b) means information isolated island. The concept of information isolated island in our paper refers to the independent of different semantic nodes can not achieve semantic inference in a homogeneous graph that connects similar semantic nodes by attribute (e.g. Figure 1(a)) or grammar (e.g. Figure 1(b)).

to its high demand of the proper semantic alignment between vision and linguistic domain. In this paper, we resolve the challenge via heterogeneous graph learning, which seamlessly integrates the intra-graph and inter-graph reasoning to bridge vision and language domain.

Current approaches mainly fall into homogeneous graph modeling with same domain (e.g. vision-to-vision graph) to mine the intra-relationship. However, one of the keys to the cognition-level problem is to excavate the inter-relationship of vision and linguistics (e.g. vision-to-answer graph) by aligning the semantic nodes between two different domains. As is shown in Figure 1, we show the difference of homogeneous graphs and heterogeneous graphs. The homogeneous graph may lead to information isolated island (dotted portion in Figure 1(a)(b)), such as vision-to-vision graph that is limited to object relationship and is not related to functionality-based information (e.g. get, pours into) from linguistics, which would hinder semantic inference. For instance, "person1 pours bottle2 into wineglass2", this sentence contains functionality-based information "pours into" which is different from the semantic visual entities such as "person1", "bottle2" and "wineglass2". It may not be enough to bridge the vision and language domain via homogeneous graph as shown in Figure 1(a)(b). However, we can connect the visual semantic node (e.g. "person1") with functional node (e.g. "pours into") from linguistics via a heterogeneous graph (Figure 1(c)), which can support proper semantic alignment between vision and linguistic domain. Benefit from the merits of the heterogeneous graph, we can seamlessly connect the inter-relationship between vision and linguistics, which can refine the reasoning path for semantic agreement. Here, we propose to use the heterogeneous graph learning for VCR task to support the visual representation being aligned with linguistics.

A heterogeneous graph module including a primal vision-to-answer heterogeneous graph (VAHG) and a dual question-to-answer heterogeneous graph (QAHG) is the core of Heterogeneous Graph Learning (HGL), which contains two steps: (1) build a heterogeneous graph and evolve the graph; (2) utilize the evolved graph to guide the answer selection. First, given the generated node representation of vision and linguistics as input, the confident weights are utilized to learn the correlation of each node. Then, to locate relevant node relationship in the heterogeneous graph conditioned by the given question and image, we utilize heterogeneous graph reasoning to get the evolved heterogeneous graph representation. Finally, given evolved graph representation, a guidance mechanism is utilized to route the correct output content.

Moreover, there exists some ambiguous semantics (e.g. rainy day) that lack of specific labels for detection and can not benefit from the labeled object bounding boxes and categories such as "person" and "dog" during training in VCR task. To solve this problem, our HGL integrates a contextual voted module (CVM) for visual scene understanding with a global perspective at the low-level features.

The key merits of our paper lie in four aspects: a) a framework called HGL is introduced to seamlessly integrate the intra-graph and inter-graph in order to bridge vision and linguistic domain, which consists of a heterogeneous graph module and a CVM; b) a heterogeneous graph module is proposed including a primal VAHG and a dual QAHG to collaborate with each other via heterogeneous graph reasoning and guidance mechanism; c) a CVM is presented to provide a new perspective for global reasoning; d) extensive experiments have demonstrated the state-of-the-art performance of our proposed HGL on three cognition-level tasks.

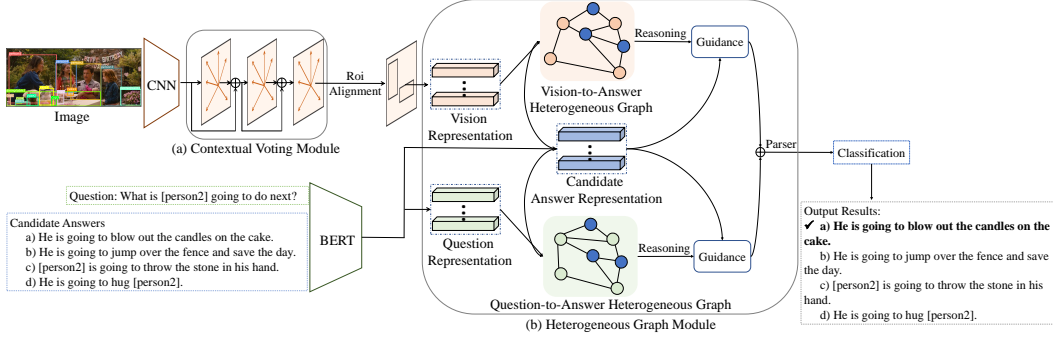

Figure 2: Overview of our HGL framework. Taking the image, question and candidate answers with four-way multiple choices as input, we use HGL to predict the right choice of candidate answers. We firstly use CNN (ResNet50 [16]) tailed with the CVM to obtain the visual representation with global reasoning. Then we utilize the shared BERT [12] to extract question representation and candidate answer representation, respectively. Then taking the three representations as the input of heterogeneous graph module, a primal VAHG module with a dual QAHG module is used to construct heterogeneous graph relationship to align semantics among vision, question and answer via heterogeneous graph reasoning and guidance, which outputs two evolved representations. The two representations are fed into a parser and classification to classify the final result.

## 2   Related Work

**Visual Comprehension** Recently, visual comprehension has made significant progress in many tasks, such as visual question answering [18, 4, 3], visual dialog [15, 37, 9] and visual question generation [25, 20]. There are mainly two aspects to methodology in the domain of visual comprehension. On the one hand, an attention-based approach was usually applied and raised its superior performance. Anderson et al. [2] presented a powerful architecture driven via bottom-up and top-down attention for image captioning and visual question answering. In multi-hop reasoning question answering task, a bi-directional attention mechanism [7] was proposed that was combined with entity graph convolutional network to obtain the relation-aware representation of nodes for entity graphs. On the other hand, a graph-based approach was developing rapidly recently, combining graph representation of questions and topic images with graph neural networks. Wu et al. [40] incorporated high-level concepts such as external knowledge into the successful CNN-RNN approach for image captioning and visual question answering. A graph-based approach [32] to model object-level relationships conditioned by the given question and image, including spatial relationship and object semantics relationship. In contrast, our proposed HGL differs in that inter-relationship is built via different domains (e.g. vision-to-answer graph) to align vision and linguistic domains.

**Graph Learning** Some researchers effort to model domain knowledge as homogeneous graph for excavating correlations among labels or objects in images, which has been proved effective in many tasks [31, 28]. Graph convolution approaches with spectral variants [5] and diffusion approaches [13] have well developed and been applied into semi-supervised node classification [24]. Some researches utilize the adjacency matrix to model the relationship of all node pairs [38, 6], while others incorporate higher-order structure inspired by simulated random walks [1, 14]. Li et al. [26] solved scene graph generation via subgraph-based model using bottom-up relationship inference of objects in images. Liang et al. [27] modeled semantic correlations via neural-symbolic graph for explicitly incorporating semantic concept hierarchy during propagation. Yang et al. [43] built a prior knowledge-guide graph for body part locations to well consider the global pose configurations. In this work, we firstly propose a heterogeneous graph learning to seamlessly integrate intra-graph and inter-graph reasoning to generate persuasive reasoning paths and bridge cross-domain semantic alignment.

## 3   Methodology

Our Heterogeneous Graph Learning (HGL) framework is shown in Figure 2. The HGL consists of a heterogeneous graph module (e.g. a primal VAHG module and a dual QAHG module) and a contextual voting module (CVM). The heterogeneous graph module is to align semantic nodes

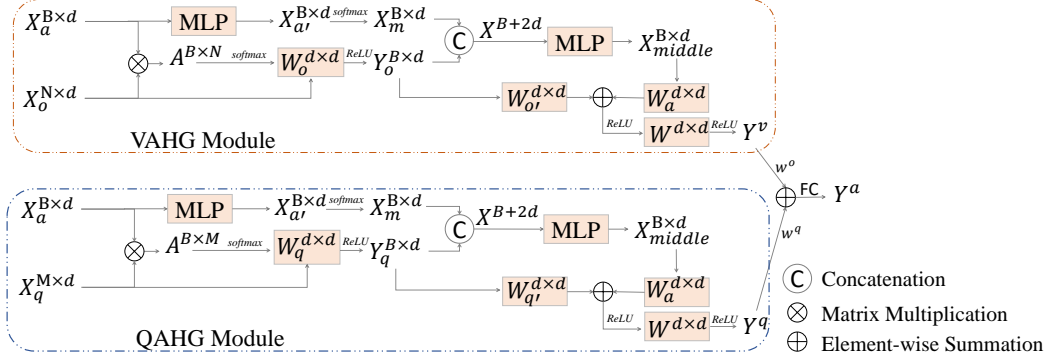

Figure 3: Implementation details of the primal VAHG module and the dual QAHG module by taking the representation of image, question and answer as inputs.

between vision and linguistic domain. The goal of CVM is to exploit long-range visual context for better global reasoning.

## 3.1 Definition

Given the object or region set $\mathcal{O} = \{o_i\}_{i=1}^N$ of an input image $\mathcal{I}$, the word set $\mathcal{Q} = \{q_i\}_{i=1}^M$ of a query and the word set $\mathcal{A} = \{a_i\}_{i=1}^B$ of the candidate answers, we seek to construct heterogeneous graph node set $\mathcal{V}^o = \{v_i^o\}_{i=1}^N$, $\mathcal{V}^q = \{v_i^q\}_{i=1}^M$ and $\mathcal{V}^a = \{v_i^a\}_{i=1}^B$, correspondingly. Each node $v_i^o \in \{\mathcal{V}^o\}$ corresponds to a visual object $o_i \in \mathcal{O}$ and the associated feature vector with $d$ dimensions indicates $\mathbf{v}_i^o \in \mathbb{R}^d$. Similarly, the associated query word vector and associated answer word vector can separately be formulated as $\mathbf{v}_i^q \in \mathbb{R}^d$ and $\mathbf{v}_i^a \in \mathbb{R}^d$. By concatenating the joint embedding $\mathbf{v}_i$ together into a matrix $\mathbf{X}$, we separately define three matrices, such as vision matrix $\mathbf{X}_o \in \mathbb{R}^{N \times d}$, query matrix $\mathbf{X}_q \in \mathbb{R}^{M \times d}$ and answer matrix $\mathbf{X}_a \in \mathbb{R}^{B \times d}$, where $N$, $M$ and $B$ denote separately the visual object number, query word number and answer word number. We define the final output of our framework as $\mathbf{Y}_p \in \mathbb{R}^4$, which is a vector with 4 dimensions according to four-way multiple choice of candidate answers as shown in Figure 2.

## 3.2 Heterogeneous Graph Learning

### 3.2.1 Vision-to-answer heterogeneous graph (VAHG) module

This module aims to align the semantics between vision and candidate answer domains via a heterogeneous graph reasoning, then generate a vision-to-answer guided representation $\mathbf{Y}^v \in \mathbb{R}^{B \times d}$ for classification via a guidance mechanism (Figure 3). We firstly introduce the reasoning of vision-to-answer heterogeneous graph, then show the guidance mechanism. We define the vision-to-answer heterogeneous reasoning as:

$$\mathbf{Y}_o = \delta(\mathbf{A}^T \mathbf{X}_o \mathbf{W}_o), \tag{1}$$

where $\mathbf{Y}_o \in \mathbb{R}^{B \times d}$ is the visual evolved representation and $\mathbf{W}_o \in \mathbb{R}^{d \times d}$ is a trainable weighted matrix. $\delta$ is a non-linear function. The $\mathbf{A} \in \mathbb{R}^{N \times B}$ is a heterogeneous adjacency weighted matrix by calculating the accumulative weights of answer nodes to vision nodes, which is formulated as:

$$A_{ij} = \frac{\exp(A_{ij}^{'})}{\sum_{ij} \exp(A_{ij}^{'})}, \quad A_{ij}^{'} = \mathbf{v}_i^{oT} \mathbf{v}_j^a, \tag{2}$$

where $A_{ij} \in \mathbf{A}$ is a scalar to indicate the correlations between $\mathbf{v}_i^o$ and $\mathbf{v}_j^a$. The $\mathbf{A} \in \mathbb{R}^{N \times B}$ is the heterogeneous adjacency weighted matrix normalized by using a softmax at each location. In this way, different heterogeneous node representation can adaptively propagate to each other.

We obtain the visual evolved representation $\mathbf{Y}_o$ via vision-to-answer heterogeneous graph reasoning (Equation (1)). Given the $\mathbf{Y}_o$ and answer matrix $\mathbf{X}_a \in \mathbb{R}^{B \times d}$ as input, we present a guidance mechanism for producing the vision-to-answer guided representation $Y^v$. We divide the guidance

mechanism into two steps. We first generate a middle representation $\mathbf{X}_{middle}$ that is enhanced by word-level attention values, then we propose a final process to generate the target representation $\mathbf{Y}^v$. The first step is formulated as following:

$$\mathbf{X}_{a'} = \mathcal{F}(\mathbf{X}_a), \tag{3}$$

$$\mathbf{x}_m = a_n \mathbf{x}_{a'}, \quad a_n = \frac{\exp(\mathbf{X}_{a'}\mathbf{W}_{a'})}{\sum_{n \in B} \exp(\mathbf{X}_{a'}\mathbf{W}_{a'})}, \tag{4}$$

$$\mathbf{X}_{middle} = f([\mathbf{X}_m, \mathbf{Y}_o]), \tag{5}$$

where $\mathcal{F}$ is a MLP used to encode the answer feature, $a_n$ is a word-level attention value by utilizing weighted product $\mathbf{W}_{a'}$ on the encoded answer feature $\mathbf{X}_{a'}$ with $B$ number words. These attention values are normalized among all word-level linguistic representation via a softmax operation, which is shown in Equation (4). Then we apply the attention value $a_n$ on $\mathbf{x}_{a'} \in \mathbf{X}_{a'}$ to produce an attention vector $\mathbf{x}_m \in \mathbf{X}_m$. We concatenate the vector $\mathbf{x}_m$ together into a matrix $\mathbf{X}_m$ that is enhanced by the attention value. After that, to combine the relationship between $\mathbf{X}_m$ and $\mathbf{Y}_o$ with $\mathbf{Y}_o$ instead of simply combining $\mathbf{Y}_o$ with $\mathbf{X}_m$, we concatenate the $\mathbf{X}_m$ with $\mathbf{Y}_o$, then utilize a MLP $f$ on the concatenated result to get a middle representation $\mathbf{X}_{middle} \in \mathbb{R}^{B \times d}$ for generating our final vision-to-answer guided representation $\mathbf{Y}^v$.

At the second step, the $\mathbf{X}_{middle}$ and the visual evolved representation $\mathbf{Y}_o$ are utilized for producing the vision-to-answer guided representation $\mathbf{Y}^v$, which is formulated as:

$$\mathbf{Y}^v = \Psi(\phi(\mathbf{Y}_o \mathbf{W}_{o'} + \mathbf{X}_{middle}\mathbf{W}_a)\mathbf{W}), \tag{6}$$

where $\mathbf{W}_{o'}$ and $\mathbf{W}_a$ are both learnable mapping matrixes to map the different embedding features into a common space to better combine the $\mathbf{Y}_o$ and $\mathbf{X}_{middle}$, and the senior weighted matrix $\mathbf{W}$ is to map the combination result into the same dimensionality. The $\Phi$ and $\phi$ are both vision-guided functions such as MLPs to get the final vision-to-answer guided representation $\mathbf{Y}^v$.

### 3.2.2 Question-to-answer heterogeneous graph (QAHG) module

In this section, we produce a question-to-answer heterogeneous graph module that is similar to VAHG. The implementation details of the QAHG module are shown in Figure 3. This module aims to support the proper semantic alignment between question and answer domains. Given the query word vector and answer word vector as input, we aim to generate the question-to-answer guided representation $\mathbf{Y}^q \in \mathbb{R}^{B \times d}$ as the final output of this module. Specifically, taking the answer vector as input, we utilize a question-to-answer heterogeneous graph reasoning to produce a query evolved representation $\mathbf{X}_q \in \mathbb{R}^{B \times d}$. Then a symmetric guidance mechanism is utilized for generating the question-to-answer guided representation $\mathbf{Y}^q$.

After getting the $\mathbf{Y}^v$ and $\mathbf{Y}^q$ from VAHG module and QAHG module, respectively, we utilize a parser at the end of the HGL framework as shown in Figure 2 to adaptively merge $\mathbf{Y}^v$ and $\mathbf{Y}^q$ to get an enhanced representation $\mathbf{Y}^a$ for final classification. The parser can be formulated as:

$$\mathbf{Y}^a = F(w^o \mathbf{Y}^v + w^q \mathbf{Y}^q). \tag{7}$$

where $w^o$ and $w^q$ are derived from the original visual feature, query feature and answer feature to calculate the importance of the task. We use a simple dot product to merge the two representations ($\mathbf{Y}^v$ and $\mathbf{Y}^q$). Then we use linear mapping function $F$ such as FC to produce the enhanced representation $\mathbf{Y}^a$ for final classification. The $w^o$ and $w^q$ can be calculated as:

$$w^o = \frac{\exp(\varphi_v([\mathbf{X}_o, \mathbf{X}_a]\mathbf{W}_{oa})}{\exp(\varphi_v([\mathbf{X}_o, \mathbf{X}_a]\mathbf{W}_{oa}) + \exp(\varphi_q([\mathbf{X}_q, \mathbf{X}_a]\mathbf{W}_{qa})}, \tag{8}$$

$$w^q = \frac{\exp(\varphi_q([\mathbf{X}_q, \mathbf{X}_a]\mathbf{W}_{qa})}{\exp(\varphi_v([\mathbf{X}_o, \mathbf{X}_a]\mathbf{W}_{oa}) + \exp(\varphi_q([\mathbf{X}_q, \mathbf{X}_a]\mathbf{W}_{qa})}, \tag{9}$$

where $\mathbf{W}_{oa}$ and $\mathbf{W}_{qa}$ are both trainable weighted matrices, and $\varphi_v$ and $\varphi_q$ indicate different MLP networks. [.] means concatenation operation.

### 3.2.3 Contextual voting module

This module aims to replenish relevant scene context into local visual objects to give the objects with a global perspective via a voting mechanism, which is formulated as:

$$y_i^l = \frac{1}{\mathcal{C}(x)} \sum_{\forall j} f\left(x_i^l, x_j^l\right) g\left(x_j^l\right),  \tag{10}$$

$$a_{x_j \rightarrow x_i} = \frac{\exp(W_n^{aT}\phi(x_i^l, x_j^l))}{\sum_{n \in \mathcal{N}} \exp(W_n^{aT}\phi(x_i^l, x_j^l))},  \tag{11}$$

$$y_i^{l+1} = a_{x_j \rightarrow x_i} y_i^l W^a + x_i^l,  \tag{12}$$

where $y_i^l$, $x_i^l$ denote the output and the input at position $i$ of $l$-th convolution layer, and $x_j^l$ denote the input at position $j$ in relevant image content. The Equation (11) represents that each output has collected global input information from relevant positions. In Equation (12), $a_{x_j \rightarrow x_i}$ denotes the learnable voting weight from position $x_j$ to $x_i$ for adaptively select relevant contextual information into local visual feature via weighted sum and element-wise product. The $W^a$ and $W_n^a$ are both trainable weights and $\phi, f, g$ are non-linear functions with conv 1×1 operation. The output of this module is the $y_i^{l+1}$, which denotes the residual visual feature maps via residual addition between input $x_i^l$ and the enhanced feature $a_{x_j \rightarrow x_i} y_i^l W^a$.

## 4 Experiments

### 4.1 Task Setup

The visual commonsense reasoning (VCR) task [44] is a new cognition-level reasoning consists of three sub-tasks: Q→A, QA→R and Q→AR. The VCR is a four-way multi-choice task, and the model must choose the correct answer from four given answer choices for a question, and then select the right rationale from four given rationale choices for that question and answer.

### 4.2 Dataset and Evaluation

We carry out extensive experiments on VCR [44] benchmark, a representative large-scale visual commonsense reasoning dataset containing a total of 290k multiple choice QA problems derived from 110k movie scenes. The dataset is officially split into a training set consisting of 80,418 images with 212,923 questions, a validation set containing 9,929 images with 26,534 questions and a test set made up of 9,557 with 25,263 queries. We follow this data partition in all experiments. The dataset is challenge because of the complex and diverse language, multiple scenes and hard inference types as mentioned in [44]. Of note, unlike many VQA datasets wherein the answer is a single word, there are more than 7.5 words for average answer length and more than 16 words for average rationale length. We strictly follow the data preprocessing and evaluation from [44] for fairly comparison.

### 4.3 Implementation

We conduct all experiments using 8 GeForce GTX TITAN XP cards on a single server. The batch size is set to 96 with 12 images on each GPU. We strictly follow the baseline [44] to utilize the ResNet-50 [16] and BERT [12] as our backbone and implement our proposed heterogeneous graph learning on it in PyTorch [33]. The hyper-parameters in training mostly follow R2C [44]. We train our model by utilizing multi-class cross entropy between the prediction and label. Each task is trained separately for question answering and answer reasoning via the same network. For all training, Adam [23] with weight decay of 0.0001 and beta of 0.9 is adopted to optimize all models. The initial learning rate is 0.0002, reducing half (×0.5) for two epochs when the validation accuracy is not increasing. We train 20 epochs for all models from scratch in an end-to-end manner. Unless otherwise noted, settings are the same for all experiments.

### 4.4 Comparison with state-of-the-art

**Quantitative results.** During this section, we show our state-of-the-art results of validation and test on VCR [44] dataset with respect to three tasks in Table 1. Note the label of the test set is not available,

| | Model | $Q \to A$ Val | Test | $QA \to R$ Val | Test | $Q \to AR$ Val | Test |
|---|---|---|---|---|---|---|---|
| | Chance | 25.0 | 25.0 | 25.0 | 25.0 | 6.2 | 6.2 |
| Text Only | BERT [12] | 53.8 | 53.9 | 64.1 | 64.5 | 34.8 | 35.0 |
| | BERT (response only) [44] | 27.6 | 27.7 | 26.3 | 26.2 | 7.6 | 7.3 |
| | ESIM+ELMo [8] | 45.8 | 45.9 | 55.0 | 55.1 | 25.3 | 25.6 |
| | LSTM+ELMo [34] | 28.1 | 28.3 | 28.7 | 28.5 | 8.3 | 8.4 |
| VQA | RevisitedVQA [19] | 39.4 | 40.5 | 34.0 | 33.7 | 13.5 | 13.8 |
| | BottomUpTopDown[2] | 42.8 | 44.1 | 25.1 | 25.1 | 10.7 | 11.0 |
| | MLB [22] | 45.5 | 46.2 | 36.1 | 36.8 | 17.0 | 17.2 |
| | MUTAN [4] | 44.4 | 45.5 | 32.0 | 32.2 | 14.6 | 14.6 |
| | R2C [44] | 63.8 | 65.1 | 67.2 | 67.3 | 43.1 | 44.0 |
| | HGL (Ours) | **69.4** | **70.1** | **70.6** | **70.8** | **49.1** | **49.8** |
| | Human | | 91.0 | | 93.0 | | 85.0 |

Table 1: Main results of validation and test dataset on VCR with respect to three tasks. Note that we do not need any extra information such as additional data or features.

and we get the test predictions by submitting our results to a public leaderboard [44]. As can be seen, our HGL achieves an overall test accuracy of 70.1% compared to 65.1% by R2C [44] on Q→A task, 70.8% compared to 67.3% on QA→R task, and 49.8% compared to 44.0% on Q→AR task, respectively. To compare with the state-of-the-art text only methods on Q→A task, our HGL performs 16.2% test accuracy improvement better than BERT [12], and even outperform ESIM+ELMo [8] by 24.2% test accuracy. Compared with the several advanced methods of VQA, our model improves around 23.9% test accuracy at least on three tasks. The superior performance further demonstrates the effectiveness of our model on the cognition-level task.

**Qualitative results.** Figure 4 shows the qualitative result of our HGL. It also shows the primal learned vision-to-answer heterogeneous graph (VAHG) and a dual learned question-to-answer heterogeneous graph (QAHG) to further demonstrate the interpretability of our HGL. As shown in Figure 4, we utilize HGL on the four candidate responses to support proper semantic alignment between vision and linguistic domains. For better comprehensive analysis, we show the weighted connections of the VAHG and QAHG according to our correct predictions on different tasks. In Figure 4(d), our VAHG successfully aligns the visual representation "person5 (brown box)" to the linguistic word "witness", and also successfully connects "person1 (red box)" with "person5" by the linguistic word "witness" to infer the right answer. Because the "person5 (brown box)" is the "witness" in this scenario. Visual representation "person1 (red box)" is connected with the emotional word "angry" to achieve a heterogeneous relationship. Based on the right answer, in Figure 4(g), our QAHG can connect the word "angry" with "gritting his teeth" for successfully reasoning the rationale. Moreover, in Figure 4(c), "feeling" from question can be aligned with the most suitable semantic word "anger" from the answer for right answer prediction, which demonstrated the effectiveness of our QAHG. These results can demonstrate that the VAHG and QAHG can really achieve proper semantic alignment between vision and linguistic domains for supporting cognition-level reasoning. The CVM is suitable to apply to visual context, because there are more information can be obtained from the visual evidence. For instance, the example of Figure 5, the felling of the "person2" must get information from visual evidence (e.g. raindrop) instead of the question to predict the right answer and reason. The arrow is pointing at raindrop and snow. More results are shown in supplementary material.

## 4.5 Ablation Studies

**The effect of CVM.** Note that our CVM learns a more enhanced object feature with a global perspective, respectively. The advantage of CVM is shown in Table 2, the CVM increases overall validation accuracy by 1.8% compared with baseline on Q→A task, 1.2% on QA→R and 3.1% on Q→AR. In Figure 5, the model w/ CVM shows the superior ability to successfully parse the semantics of rain and snow in the image for better understand the rainy/snowy scene by highlighting the relevant context as indicated by the red arrows in Figure 5(b).

| Model | $Q \to A$ | $QA \to R$ | $Q \to AR$ |
|---|---|---|---|
| Baseline | 63.8 | 67.2 | 43.1 |
| Baseline w/ CVM | 65.6 | 68.4 | 45.4 |
| Baseline w/ QAHG | 66.1 | 68.2 | 45.8 |
| Baseline w/ VAHG | 66.4 | 69.1 | 46.4 |
| HGL w/o CVM | 68.4 | 69.7 | 48.3 |
| HGL w/o QAHG | 67.8 | 69.9 | 48.2 |
| HGL w/o VAHG | 68.0 | 68.8 | 48.0 |
| HGL | **69.4** | **70.6** | **49.1** |

Table 2: Ablation studies for our HGL on three tasks over the validation set.

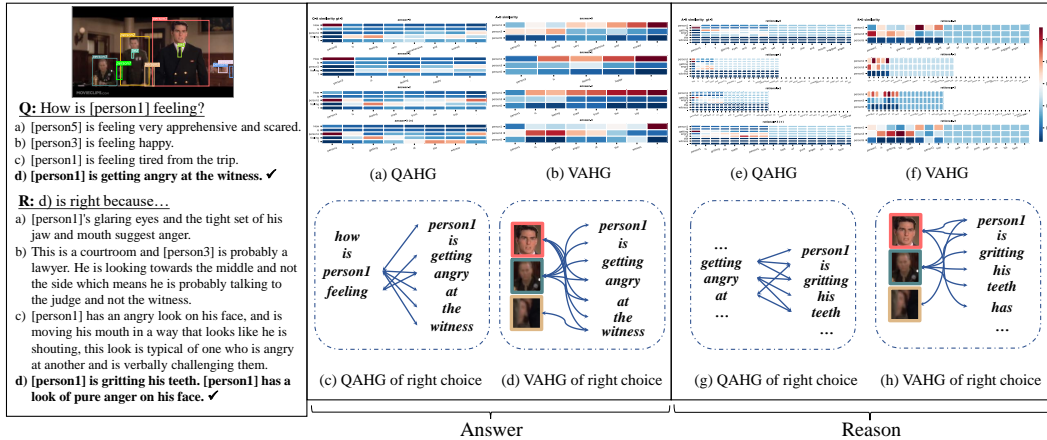

Figure 4: Qualitative results of VAHG and QAHG. (a)(b)(e)(f) are the learned VAHG and QAHG of four-way multiple choices on answer task and reason task, respectively. (c)(d)(g)(h) show the weighted connection of VAHG and QAHG according to the prediction from four choices. The predicted result is shown as **bold** font, and the ground truth (GT) is shown as ✓. Please zoom in the colored PDF version of this paper for more details.

**The effect of QAHG.** The effect of QAHG module is apparently to boost the validation accuracy by around 1.0% from baseline on all tasks. In Figure 4(c), the "feeling" from the question is connected with the "getting angry" from the right answer choice, and the "getting angry" from the question is connected with "gritting his teeth" from the rationale in Figure 4(g), which demonstrates the effectiveness of QAHG that can generate persuasive reasoning paths.

**The effect of VAHG.** We analyze the effect of VAHG on VCR. The VAHG module can promote the baseline by 2.6% (Q→A), 2.1% (QA→R) and 3.3% (Q→AR) accuracy. In Figure 4(h), the visual representation "person1 (red box)" is semantically aligned to the word "person1". The visual representation "person5 (brown box)" is semantically aligned to the word "witness" in Figure 4(d), and "person1 (red box)" and "person5 (brown box)" are connected by the word "witness". Based on these relationships, the HGL can refine reasoning paths for semantic agreement.

**HGL w/o CVM.** As can be seen in Table 2, the effect of combination between VAHG module and QAHG module can reach a high performance to 68.4%, 69.7% and 48.3%, which can demonstrate the effectiveness of building VAHG and QAHG to bridge vision and language domains.

**HGL w/o QAHG.** The proposed CVM collaborated with VAHG module is evaluated on three tasks and performs 67.8%, 69.9% and 48.2%, correspondingly. It can support the feasibility of incorporating CVM with VAHG module.

**HGL w/o VAHG.** The ability of CVM+QAHG is shown in Table 2, which gets great scores on overall tasks as validating the availability of the combination.

**Q:** What if [person2] fell?
**A:** Person2 would get wet.
**R:** Preson2 is surrounded by water.

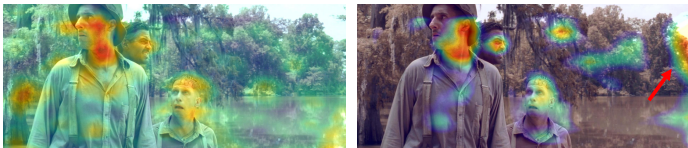

**Q:** Is it snowing outside?
**A:** Yes, it is snowing.
**R:** [person4] is dressed in a hat, scarf and a big jacket, his hat and shoulders are covered in white snowflakes.

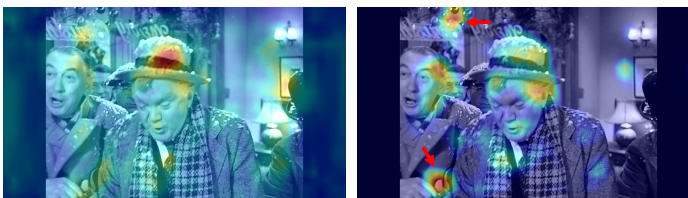

(a)           (b)

Figure 5: Qualitative results of our CVM. (a) The model w/o CVM. (b) The model w/ CVM.

## 5 Conclusion

In this paper, we proposed a novel heterogeneous graph learning framework called HGL for seamlessly integrating the intra-graph and inter-graph reasoning in order to bridge the proper semantic alignment between vision and linguistic domains, which contains a dual heterogeneous graph module including a vision-to-answer heterogeneous graph module and a question-to-answer heterogeneous graph module. Furthermore, the HGL integrates a contextual voting module to exploit long-range visual context for better global reasoning.

## 6 Acknowledgements

This work was supported in part by the National Natural Science Foundation of China (NSFC) under Grant No.1811461, in part by the National Natural Science Foundation of China (NSFC) under Grant No.61976233, and in part by the Natural Science Foundation of Guangdong Province, China under Grant No.2018B030312002.

## Footnotes

*Corresponding author is Xiaodan Liang

[2]Our code is released in https://github.com/yuweijiang/HGL-pytorch

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
