[Supplementary Material · supplementary-nips1586.pdf]

# Supplementary Material
# Heterogeneous Graph Learning for Visual Commonsense Reasoning

The supplementary material presents more experimental results of our heterogeneous graph learning framework (HGL) on VCR [1] dataset.

There are three section in our supplementary material to show our visualization and interpretation.

Sec. 1 provides the results of visual commonsense reasoning in VCR, as shown in Figure 1 and Figure 2.

Sec. 2 provides a pipeline of the interpretative visualization as shown in Figure 3.

Sec. 3 aims to show the details of the proposed modules to demonstrate the effectiveness and interpretation of our VAHG, QAHG and CVM, which is shown in Figure 6 4 5 9 7 8 12 10 11 15 13 14 18 16 17,.

To fully demonstrate the effectiveness and interpretability of our proposed modules, we provide five examples (Figure 6 4 5 9 7 8 12 10 11 15 13 14 18 16 17) to show the learned vision-to-answer heterogeneous graph and question-to-answer heterogeneous graph, the visualization of heatmap to demonstrate the effetiveness of proposed contextual voting module.

## 1    Prediction Results

This section aims to present the prediction results of our HGL on VCR. Not only do we show the right results, but also we display our wrong prediction to better analyse our method, which is shown in Figure 1 2. It can be seen that our HGL outperforms well on many situations, while it may not well fit for the human emontion inference such as shown in the second example of Figure 2.

## 2    Interpretation Pipleline

This section presents a whole pipleine of our interpretation visualization, which is provided by Figure 3. There three ways of our visualization.

First, we utilize the correlative matrix to display the learned heterogeneous graph edges.

Second, we show a specific connection for VAHG and QAHG of the right prediction of our HGL for better understanding the semantic alignment between vision and linguistic domains. Note that the line thickness represents the weight.

Third, we provide a heatmap visualization to perform the highlight area in the visual scene conditioned by different tasks to further comprehend the contextual voting module.

**Q: What is [person1] doing?**
**Answers:**
a) [person1] is telling [person 3] to come along with her because it is time to cut [cake1].
b) She is going to pour [person 9] on [person 10].
**c) She is placing the plate with [cake1] on the table. ✓**
d) [person 1] is beckoning [person 10] to dance with her.

**Reasons:**
a) She has a cupboard on the table and is applying paste on it.
**b) The plate with the cake is still above the table, and she is holding it. It is customary to place the cake in front of the person celebrating their birthday. ✓**
c) She looks like she is preparing something for the man.
d) She is younger than the men and sometimes men have women do work like bringing the food and there is a table of snacks behind her.

**Q: What is [person2] going to do next?**
**Answers:**
**a) He is going to blow out the candles on the cake. ✓**
b) He is going to jump over the fence and save the day.
c) [person2] is going to throw the stone in his hand.
d) He is going to hug [person2].

**Reasons:**
**a) On someone' s birthday , they blow out the candles on their cake and make a wish. ✓**
b) It is typical for someone to blow out the candles on their birthday cake.
c) The candle is lit so there would be a fire is the candle was knocked over.
d) The candles would easy burn the table cloth material.

**Q: Why is [person1] looking down?**
**Answers:**
a) She is checking the store' s inventory.
b) [person1] is looking down because she' s reading.
c) There are a bunch of dishes in her [sink1].
**d) So she doesn' t spill any liquid. ✓**

**Reasons:**
**a) She is pouring [bottle2] into [wineglass2]. ✓**
b) There is a chance [person1] will get in the pool and she does not want to get they wet.
c) [person2] looks very uneasy and maybe she is trying to keep information from the cop and is afraid that [person1] will spill.
d) There is a body of water behind her.

**Q: What is [person1] doing?**
**Answers:**
a) [person1] is leading a protest.
b) [person1] is looking for a section in the bookstore.
**c) [person1] is pouring some liquor from [bottle2] into [wineglass2]. ✓**
d) [person1] is waiting on [person3].

**Reasons:**
a) [wineglass2] is a type of wineglass that is used to drink wine out of, it is half - way full and she is holding it implying she is drinking from it.
b) [wineglass2] holds [person3]'s beverage. She has it sitting in front of her on the table for when she gets thirsty.
c) You can tell that [person2] is offering [wineglass2] to her friend and [wineglass2] is a bottle of alcohol.
**d) She has the bottle tipped to the side. ✓**

Figure 1: Qualitative results of our HGL. The predicted result is shown as **bold** font, and the ground truth (GT) is shown as ✓. Please zoom in the colored PDF version of this paper for more details.

# 3 Interpretation of Modules

In this section, we display five examples to show the effectiveness of each module that is applied into different scenes in detail. Note that each example contains three types of visualization, such as the visualization of vision-to-answer heterogeneous graph, the visualization of question-to-answer heterogeneous graph and the visualization of contextual voting module.

The first example is shown in Figure 5, Fig 4 and Figure 6 to demonstrate the effectiveness and interpretation of VAHG, QAHG and CVM in two tasks, respectively.

The second example is shown in Figure 8, Fig 7 and Figure 9 to demonstrate the effectiveness and interpretation of VAHG, QAHG and CVM in two tasks, respectively.

The third example is shown in Figure 11, Fig 10 and Figure 12 to demonstrate the effectiveness and interpretation of VAHG, QAHG and CVM in two tasks, respectively.

The fourth example is shown in Figure 14, Fig 13 and Figure 15 to demonstrate the effectiveness and interpretation of VAHG, QAHG and CVM in two tasks, respectively.

The fifth example is shown in Figure 17, Fig 16 and Figure 18 to demonstrate the effectiveness and interpretation of VAHG, QAHG and CVM in two tasks, respectively.

**Q: Where is [person4] currently ?**
Answers:
a) [person4] is in prison.
b) [person4] is in a poultry processing plant.
c) [person4] is on a dance floor.
**d) [person4] is in a board room meeting.** ✓

Reasons:
a) There ' s a floral tablecloth on the table which is not typically seen in conference rooms.
b) There is a board directly behind [person4] with calculations on it .
c) There are a lot of people all in business clothes which is typical of announcements and meetings for businesses.
d) **[person4] is seated at a long conference table with [person1, person2, person3] , they appear to be talking , there are papers and clipboards on the table.** ✓

**Q: What are [person1, person 2, person 3, person 4] meeting about ?**
Answers:
**a) They are in a meeting about art.** ✓
b) They are discussing the merits of free trade versus protectionism.
c) All the people that helped them in their career.
d) [person1, person 2, person 3, person 4] are listing to the testimony happening in front of them.

Reasons
a) They are dressed up and there is a portrait on an easel behind them.
b) There are paintings with special lighting hanging on the walls behind them.
**c) On the walls around the building there are various artworks, suggesting that art is part of the company portfolio.** ✓
d) The art work is rudimentary and in crayon.

**Q: What were people [person1, person2, person3] , and [person4] doing originally ?**
Answers:
a) They were staring at the people person addressing them.
b) Everyone was going to a church.
c) People were previously dancing.
**d) They were eating a meal.** ✓

Reasons:
a) The room consists of long tables filled with numerous people that appear to be sitting down to eat a meal .
**b) The table is spread out with food , and drink and they are all seated . this indicates they were eating at some point .** ✓
c) There are several other people that are all clutching their stomachs . there is something dripping from the top of the men seated in front of [person1, person2, person3] that is the same color indicating they just vomited the same meal .
d) There are plates on food on the table in front of each of them.

**Q: How is [person2] feeling ?**
Answers
a) [**person2**] is feeling lonely and concerned for his safety .
b) He is angry.
**c) He is feeling scared** .
d) He is surprised. ✓

Reasons:
a) His eyes are closed and his face hints at controlled surprise.
b) His mouth is agape and is staring in the direction of [**person2**].
c) He appears to be looking up as if he was startled by something.
**d) His eyes are wide open and his eyebrows are furrowed, this indicates surprise in humans .** ✓

Figure 2: Qualitative results of our HGL. The predicted result is shown as **bold** font, and the ground truth (GT) is shown as ✓. Please zoom in the colored PDF version of this paper for more details.

**(a)**

**Q:** Where is [person2] walking towards?

a) [person2] is walking to the glass office at the end of the walkway.
b) **[person2] is walking into the building that [person1] [person3] walking out of. √**
c) [person2] is walking to work.
d) [person2] is walking to the other side of the pool.

**R:** b) is right because...

a) Is it is common courtesy when enter a building to let people exit or enter before you yourself enter.
b) [person2] is walking into the doorway, so it is obvious that this is where [person2] is going.
c) [person2] **is facing the building and heading upwards on the stairs towards the building. √**
d) [person1] [person3] are walking down a hallway with multiple doors that have numbers on them like an apartment complex would.

Image
Answer Heatmap
Reason Heatmap

QAHG  VAHG  QAHG  VAHG

**(b)**

**Q:** What is [person4] doing?

a) [person4] is attempting to remove a panel.
b) [person4] is handing something to [person3].
c) [person4] is watching [person1] intently.
d) **[person4] is painting the house. √**

**R:** d) is right because...

a) [person4] has something that looks like an eraser in his hand and is rubbing the picture with it.
b) He is holding a canvas and has possibility painted something on it. He is talking to the other two people and is possibly telling them about his painting.
c) [person4] is bending over while holding a hoe int he garden soil.
d) **[person4] is holding a bucket and the part of the house in front of him is partially painted.√**

Image
Answer Heatmap
Reason Heatmap

QAHG  VAHG  QAHG  VAHG

**(c)**

**Q:** What if [person2] fell?

a) Everyone else would fall back.
b) [person2] might drop [backpack1].
c) The forklift would tip over.
d) **[person2] would get wet. √**

**R:** d) is right because...

a) Almost everyone around him is holding an umbrella, but [person2] does not seem to have brought one today.
b) [person2] wouldn't be able to get up the stairs.
c) Cameras are not usually designed to get wet, so the camera would be ruined by the water.
d) **[person2] is surrounded by water. √**

Image
Answer Heatmap
Reason Heatmap

QAHG  VAHG  QAHG  VAHG

Figure 3: The pipeline of the interpretative visualization. The predicted result is shown as **bold** font, and the ground truth (GT) is shown as √. Please zoom in the colored PDF version of this paper for more details.

**Q:** Why are [ person3 person4 ] attacking [person2] ?

a) **[person2] was trying to kill [ person3 person4 ] and it did not work.** ✓
b) The bikers are mad at [person2] and want to hurt them.
c) [ person3 person4 ] are policeman, it appears [person2] was in a fight.
d) They want him to leave.

(a) QAHG

(b) QAHG of Prediction

(c) VAHG

(d) VAHG of Prediction

Figure 4: The first example in answer task. The predicted result is shown as **bold** font, and the ground truth (GT) is shown as ✓. Note that the line thickness represents weight, more thick line denotes more important connection according to more high value in the correlative matrixs.

**R:** a) is right because…

a) **[ person3 person4 ] now have the upper hand.** ✓
b) They are bloodthirsty warriors and watching men kill each other fulfills their psychotic desires.
c) Many women are assaulted sexually each day all over the world it is sadly a common reason to attack a woman.
d) [person2] was running around and being loud so they bound her hands and feet and gagged her.

(a) QAHG

(b) QAHG of Prediction

(c) VAHG

(d) VAHG of Prediction

Figure 5: The first example in reason task. The predicted result is shown as **bold** font, and the ground truth (GT) is shown as ✓. Note that the line thickness represents weight, more thick line denotes more important connection according to more high value in the correlative matrixs.

(a) Heatmap of answering

(b) Heatmap of reasoning

Figure 6: Visualization of CVM to the first example in two tasks.

**Q:** What is [person1] thinking ?

a)She's thinking that [person3] is very interesting.
**b)[person1] is wondering why [person4] is talking to her.** ✓
c)She would like to leave.
d)She is probably wondering why there are is a dance party in the restaurant.

(a) QAHG

(b) QAHG of Prediction

(c) VAHG

(d) VAHG of Prediction

Figure 7: The second example in answer task. The predicted result is shown as **bold** font, and the ground truth (GT) is shown as ✓. Note that the line thickness represents weight, more thick line denotes more important connection according to more high value in the correlative matrixs.

**R:** b) is right because…

a)[person1] is turned around and is staring at [person4].
**b)[person1] appears to look unsocial and very unfriendly. It appears as if [person1] does not want to hear
what [person4] has to say.** ✓
c)[person1] is still talking to the group, but is looking at [person4].
d)[person1] is looking at [person4] with a curious expression.

(a) QAHG

(b) QAHG of Prediction

(c) VAHG

(d) VAHG of Prediction

Figure 8: The second example in reason task. The predicted result is shown as **bold** font, and the
ground truth (GT) is shown as ✓. Note that the line thickness represents weight, more thick line
denotes more important connection according to more high value in the correlative matrixs.

(a) Heatmap of answering

(b) Heatmap of reasoning

Figure 9: Visualization of CVM to the second example in two tasks.

**Q:** Where is [person2] walking towards?

a) [person2] is walking to the glass office at the end of the walkway.
**b) [person2] is walking into the building that [ person1 person3 ] are walking out of.** ✔
c) [person2] is walking to work.
d) [person2] is walking to the other side of the pool.

(a) QAHG

(b) QAHG of Prediction

(c) VAHG

(d) VAHG of Prediction

Figure 10: The third example in answer task. The predicted result is shown as **bold** font, and the ground truth (GT) is shown as ✓. Note that the line thickness represents weight, more thick line denotes more important connection according to more high value in the correlative matrixs.

**R:** b) is right because…

a) Is it is common courtesy when enter a building to let people exit or enter before you yourself enter.
b) [person2] is walking into the doorway, so it is obvious that this is where [person2] is going.
c) **[person2] is facing the building and heading upwards on the stairs towards the building.** ✔
d) [ person1 person3 ] are walking down a hallway with multiple doors that have numbers on them like an apartment complex would.

(a) QAHG

(b) QAHG of Prediction

(c) VAHG

(d) VAHG of Prediction

Figure 11: The third example in reason task. The predicted result is shown as **bold** font, and the ground truth (GT) is shown as ✓. Note that the line thickness represents weight, more thick line denotes more important connection according to more high value in the correlative matrixs.

(a) Heatmap of answering

(b) Heatmap of reasoning

Figure 12: Visualization of CVM to the third example in two tasks.

**Q:** Why is [person2] gesturing with his hands?

a) [person2] is explaining how to dance.
**b) [person2] is having an argument at the dinner table. ✓**
c) [person2] is trying to explain the rock to [person1].
d) [person2] is telling the crowd of people behind [ person1 person4 ] to be quiet.

(a) QAHG

(b) QAHG of Prediction

(c) VAHG

(d) VAHG of Prediction

Figure 13: The fourth example in answer task. The predicted result is shown as **bold** font, and the ground truth (GT) is shown as ✓. Note that the line thickness represents weight, more thick line denotes more important connection according to more high value in the correlative matrixs.

**R:** b) is right because…

a) [person2] is at the dinner table and calling other people for dinner.
**b) Everyone is at a dinner table and [person2] is making an exasperated expression while talking.** ✓
c) Both [ person2 person4 ] are sitting at the table and [person2] is gesturing.
d) Some people talk wildly with their hands.

(a) QAHG

(b) QAHG of Prediction

(c) VAHG

(d) VAHG of Prediction

Figure 14: The fourth example in reason task. The predicted result is shown as **bold** font, and the ground truth (GT) is shown as ✓. Note that the line thickness represents weight, more thick line denotes more important connection according to more high value in the correlative matrixs.

(a) Heatmap of answering

(b) Heatmap of reasoning

Figure 15: Visualization of CVM to the fourth example in two tasks.

**Q:** Why is [ person1 person2 ] staring at [person3]?

    a) [person3] is their leader.
    **b) [person3] should be sitting at the dining table.** ✓
    c) [ person1 person2 ] are there to arrest [person3].
    d) They have been waiting for [person3] to return with their goods.

(a) QAHG

(b) QAHG of Prediction

(c) VAHG

(d) VAHG of Prediction

Figure 16: The fifth example in answer task. The predicted result is shown as **bold** font, and the ground truth (GT) is shown as ✓. Note that the line thickness represents weight, more thick line denotes more important connection according to more high value in the correlative matrixs.

**R:** b) is right because…

a) [person3] is somewhat close to [diningtable3] so [person3] could have arrived on [diningtable3].
b) **[person3] is at the window instead of sitting at the dining table.** ✓
c) [ person1 person2 ], and [person3] are turned around to see something but it isn't [diningtable2]. [diningtable2] is a chair and not very interesting.
d) We see only one chair. probably others want [person3] to use chair.

(a) QAHG

(b) QAHG of Prediction

(c) VAHG

(d) VAHG of Prediction

Figure 17: The fifth example in reason task. The predicted result is shown as **bold** font, and the ground truth (GT) is shown as ✓. Note that the line thickness represents weight, more thick line denotes more important connection according to more high value in the correlative matrixs.

(a) Heatmap of answering

(b) Heatmap of reasoning

Figure 18: Visualization of CVM to the fifth example in two tasks.

# References

[1] R. Zellers, Y. Bisk, A. Farhadi, and Y. Choi. From recognition to cognition: Visual commonsense reasoning. *arXiv preprint arXiv:1811.10830*, 2018.