[Reviews · NeurIPS 2019]

Reviewer 1



Originality: The VCR task is a novel task (proposed by Zellers et al, CVPR19). The proposed HGL framework for this interesting task is novel and interesting. The paper applies the HGL framework on top of the baseline model (R2C from Zellers et al., CVPR19) and shows significant improvements. The paper compares other existing graph learning approaches. The main difference between the proposed approach and other graph learning approaches is the heterogeneous nature (across domains – vision and language) of the graph learning framework. Quality: The paper does a good job of evaluating the propsed approach and its ablations. The visualizations reported in Fig. 4 are also useful. However, I have some concerns -- 1. Fig. 4 a,b,e,f – the intersity of the visualizations is counter-intuitive for some cases, such as “feeling” and “anger” for Fig. 4(a), “anger” and “angry” for Fig. 4(e), “person5” and “witness” for Fig. 4(b), “person1” and angry for Fig. 4(e). Could authors please comment on this? 2. In Fig. 5(a), the arrow is pointing at sky, instead of pointing at water in the water body. It’s not clear how CVM is helpful here. Could authors please comment on this? 3. It is becoming concerning in vision and language literature that models rely on priors (although language) more than grounding their evidence on the visual context. Could authors please comment on whether the use of Contextual Voting Module to incorporate priors could lead to models relying on context more than the true visual evidence? It could be useful to analyze the instances where CVM helps and see if any such undesired prior learning is happening. 4. Why is Y_o concatenated with X_m in Eq. 5 when Y_o is getting combined with X_middle anyways in Eq. 6? How much does the concatenation in Eq. 5 affect the final results? Clarity: The paper is clear for the most part. I have a few clarification questions / comments -- 1. The concept of information isolated island on L41-42 is not explained. 2. There are some typos in L140-153, such as “shown in Eq. 5” à “shown in Eq. 4”, “f_v” à “f”, inconsistent notations for psi in Eq. 6 and L152. 3. Not all the notations in Eq.8 and Eq.9 are explained. Also should it be W_{oa} and W_{qa}, instead of W_oa and W_qa? 4. What is y_{i}^{rl} in Eq. 12. Specifically, it’s not clear what “r” refers to and how is that related with y_{i}^{l}. 5. It would be good for this paper to describe briefly the three tasks: Q à A, QA àR, Q à AR. 6. L247 – how is the information about tags associated with the bounding boxes in the image, such as “person1” tag associated with red box, “person5” tag associated with brown box fed to the model? Significance: The task of VCR is an interesting, relatively new and useful task. The paper proposed an interesting, interpretable, novel approach for VCR that also significantly advances the state-of-the-art. I think the proposed work could be of high significance for pushing the research in VCR. --- Post-rebuttal comments --- The authors have responded to all my concerns. Most of the responses are satisfactory, except the following two -- The concern about visualizations in Fig 4 was that the intensities are too low for the instances pointed out in the review. The two words in the each of the instance should be aligned with each other (as also pointed out by the authors in the rebuttal), however the intensities are otherwise. It would be good for the authors to look more into this and clarify this in the camera-ready version. Regarding the CVM – I see the usefulness of it in situations where context is needed. But I am worried about situations when context is not needed but CVM can potentially hurt because it is easier for the model to predict the answer by the prior knowledge based on the context. For instance, predicting “cooking” for “what is the person doing” because the task is being performed by a woman and CVM is using the context of woman. Given the novelty and significance of the proposed approach, I would recommend acceptance of this paper.

Reviewer 2



Post rebuttal: Thank you very much for your response it has addressed my questions and happy to increase my score. The paper proposes a new model for VQA which computes bi-directional attention over the image + answers as well as over question words + answers. Related ideas have mainly been explored in reading comprehension (e.g. BiDAF [1]) but also in VQA [2,3,4]. The paper discusses the model in the context of a new dataset called VCR, for visual commonsense reasoning. The paper is well-structured, although there is room for further improvement of writing and clarity, especially of the model section. Good visualizations and ablations studies are provided. My main concern however is the fact that results are only provided for VCR, without experimenting on more common VQA tasks such as VQA1/2. Since the key aspect of the model is the bidirectional attention between the question/image and the words in the answer sentences in VCR, it is not clear to me whether the model will work well compared to existing approaches for standard VQA tasks where the answers are usually 1-2 words. Model - The description of the Vision-to-answer heterogeneous graph module (section 3.2.1) was a bit hard for me to follow - I didn’t fully understand the structure of that module from the description, especially the part about the guidance mechanism and the word-level attention. It might help to give a more intuitive / higher-level explanation of the goal of each computation. - I like the high-level idea of the model as discussed in the introduction, of computing graphs both over the image and the question to find more semantic relations between the objects, as illustrated in figure 2 (e.g. person - (pours) -> bottle). However, I don’t fully understand where the visual or question modules infer these graphs in practice? If I understand correctly, it seems that the adjacency matrix used (end of page 4) simply captures the similarity in vector space between each of the objects/question words and each of the answers, rather than computing relations between the different objects themselves. Is this covered by a different module? The visualizations in page 8 and supplementary indeed show only attention between objects and answers, or question words and answers, but don’t indicate spatial/semantic relations in between the objects / question words themselves. - I’m wondering about the necessity of having an independent contextual voted module (CVM) that is meant to capture global information about the image rather than specific objects. Wouldn’t it be possible to add additional “context” node/s to the same graph instead of having two separated modules? The “objects” in the graph don’t necessarily have to be physical concrete objects, but rather, any visual “entity” in the image (any visual features). Merging both modules into one graph would make the model in my opinion more unified. - I also don’t understand some of the explanation about CVM. The paper says its goal is “to exploit long-range visual context for better global reasoning”. What does long-range visual context mean? The paper doesn’t explain that at any point - a concrete example may help clarify that point. Experiments - The paper provides experiments only over a new dataset for visual commonsense reasoning called VCR. It would be good if experiments could be provided also for more standard datasets, such as VQA1/2, or the newer VQA-CP, to allow better comparison to many existing approaches for visual question answering. - I’m also wondering whether or not the model could perform well over standard VQA tasks - it seems that it mainly builds on the fact that the answers are sentences and can relate in different ways to each of the objects or question words. It therefore would be very important to see how the model performs on more standard VQA tasks where the answers are usually much shorter - about 1-2 words. - Most VQA models so far tend to use word embedding + LSTM to represent language - Could you please perform an ablation experiment using such approach? While it’s very likely that BERT helps improve performance, and indeed the baselines for VCR use BERT, I believe it is important to also have results based on the more currently common approach (LSTM) to allow comparison to most existing VQA approaches. Clarity - (Page 1): In the first paragraph, the paper discusses vision-and-language tasks, and then provides a couple of examples: including (1) bottom-up attention and BERT. (“One type is to explore a powerful end-to-end network. Jacob et al. have introduced a powerful end-to-end network named BERT”). While I understand that BERT is mentioned here because the model later on uses it, I personally think it doesn’t make a lot of sense to give BERT as an example of a vision-and-language model. I think would be better instead to mention it in the Related Work section, briefly discussing BERT’s merits and how the new model uses it. - Some descriptions are a little bit repetitive (same sentences or explanations repeat verbatim in multiple places, e.g. “HGL integrate a contextual voting module / CVM to exploit long-range visual context for better global reasoning” which returns 3 times, and I noticed a couple more such cases). - (Page 4) It seems that the title “Task Definition” is probably not really describing the content of that subsection - maybe “graph construction” would be a better fit? - (Page 4) On line 131 it says that the “correlations” between the vectors are computed. I believe this means e.g. a dot product between them? It would probably be clearer to call it that way, or alternatively give a more precise definition of how the correlation is computed. - (Supplementary page 2) there are 5 duplicated sentences for each of the five examples - would be better to replace them with one paragraph that says something along the lines of “In the following, we provide multiple visualizations (figures 1-5) to demonstrate the effectiveness and interpretation of…” rather than repeating that same sentence five times. - Increasing the font size inside the figures in the paper and supplementary would be really helpful. - A few specific typos I noticed: *Page 2: boottle -> bottle *Page 2: “which consists a” -> consists of a *Page 3: “In multi-top reasoning” -> multi-hop *Page 3: “The CVM is to exploit” - missing word, maybe cvm’s goal is to *Page 3: “and applied into” -> and been applied to *Page 4: “we divide guidance mechanism” -> the guidance *Page 5: “dot production” -> dot product *Page 5: “use a simple dot production to merging” -> to merge *Page 5: “to produce the for final” -> missing word after the *Page 6: “local visual local object feature” -> mabe “local visual feature” *Page 6: “cars” -> cards *Page 7: “we show the weighted connect of” -> probably connections? *Page 8: the HGL integrate -> integrates [1] Seo, Minjoon, Aniruddha Kembhavi, Ali Farhadi, and Hannaneh Hajishirzi. "Bidirectional attention flow for machine comprehension." arXiv preprint arXiv:1611.01603 (2016). [2] Lu, Jiasen, Jianwei Yang, Dhruv Batra, and Devi Parikh. "Hierarchical question-image co-attention for visual question answering." In Advances In Neural Information Processing Systems, pp. 289-297. 2016. [3] Hudson, Drew A., and Christopher D. Manning. "Compositional attention networks for machine reasoning." International Conference on Learning Representations, 2018 [4] Nam, Hyeonseob, Jung-Woo Ha, and Jeonghee Kim. "Dual attention networks for multimodal reasoning and matching." In Proceedings of the IEEE Conference on Computer Vision and Pattern Recognition, pp. 299-307. 2017.

Reviewer 3



*Originality* The approach is novel and an interesting way to combine structured information in the sentence and the image. It could be of use for other vision and language tasks, such as visual reasoning. *Quality* The description of the model looks sound. However, it’s unclear whether the graphs are latent or provided during training. Clarifying the inputs and outputs of the model during training and during inference would be useful. A discussion of this approach’s possible applications to other vision and language tasks would be appreciated. A more comprehensive qualitative analysis, e.g., more (larger) examples would improve the paper, especially examples and analyses that demonstrate the interpretability aspect of this approach. *Clarity* The abstract and introduction assume understanding of highly specific terms, e.g., heterogenous vs. homogenous graphs, intra- vs. inter-graph, and several acronyms that might not be familiar to those not working on VCR. However, most of it was made clearer in the approach section. But to get readers interested, this should be edited. There seems to be an issue with the citations (some peoples’ first names are being cited instead of their last, like Jacob et al. in L24). *Significance* This approach could be of use to researchers working on other vision and language tasks. It achieves SOTA on all three tasks of VCR, and provides interpretable results.

[Author Response · NeurIPS 2019]

| Model | $Q \to A$ | | $QA \to R$ | | $Q \to AR$ | |
|---|---|---|---|---|---|---|
| | GloVe | BERT | GloVe | BERT | GloVe | BERT |
| BottomUpTopDown [2] | 42.8 | 62.3 | 25.1 | 63.0 | 10.7 | 39.6 |
| R2C [46] | 46.4 | 63.8 | 38.3 | 67.2 | 18.3 | 43.1 |
| **HGL (ours)** | **54.1** | **69.4** | **42.7** | **70.6** | **25.1** | **49.1** |

Table 1: Results on the VCR validation set. While BERT helps the performance of these baselines, our model still performs the best.

**To All:** We thank all reviewers for the constructive and insightful comments and address the comments as follows. We appreciate the fact that all reviewers give positive evaluation results about our HGL. We promise to revise typos and grammatical mistakes and polish our paper in the revised version based on these valuable suggestions.

**Q1: Clarification on results of Fig.4 and Fig.5 (R1).** In Fig.4(a), "feeling" from question can be aligned with the most suitable semantic word "anger" from the answer for right answer prediction, which demonstrated the effectiveness of our QAHG. In Fig.4(e), the visual representation "person5" is aligned with the linguistic word "witness" correctly via VAHG, because "person5" is the "witness" in this scenario. Visual representation "person1" (tag associated with the red box is fed to model) is connected with the emotional word "angry" to achieve a heterogeneous relationship. In Fig.5, the person would fell wet because of rain in the scenario. The arrow is pointing at raindrop.

**Q2: Comment on CVM applied to vision (R1).** The CVM is suitable to apply to visual context, because there are more information can be obtained from the visual evidence. For instance, the first example of Figure 1 from supplementary material, the motivation of person1 must get information from visual evidence (e.g. cake1, table and background) instead of the question to predict the right answer and reason.

**Q3: Discussions on the affection of concatenation in Eq.5 (R1).** To combine the relationship between $X_m$ and $Y_o$ with $Y_o$ instead of simply combining $Y_o$ with $X_m$, we need the concatenation operation in Eq.5. We did experiments to analyze the affection and found the concatenation can improve ~0.4% accuracy (69 vs 69.4) on Q->A task over VCR validation set. The experiment showed that the concatenation operation can help the model to further understand the heterogeneous relationship for predicting correctly.

**Q4: More experiments on common VQA datasets (R2).** We trained our HGL on VQA2 dataset following the same experimental setup [2], and evaluated our HGL on VQA2 validation dataset compared with common VQA approach such as BottomUpTopDown [2]. The BottomUpTopDown [2] achieved 63.2 accuracy across all question types and the HGL achieved 65.3 accuracy, which showed our proposed model can also be transferred to VQA task and perform well.

**Q5: Ablation experiments between BERT and common VQA backbone (e.g. GloVe) (R2).** In Table 1, the BERT indeed helped the performance of HGL (54.1→69.4, 42.7→70.6, 25.1→49.1), our HGL still got SOTA performance compared with common VQA approach such as BottomUpTopDown [2], and R2C [46] on VCR validation set.

**Q6: Clarification on CVM and the possibility to take "visual context" as graph nodes (R2).** The CVM aims to capture ambiguous semantic context (e.g. rainy/snowy weather) that lack of specific labels for detection and can not benefit from the labeled object grounding boxes and categories such as "person" and "dog" during training. The long-range visual context means spacial visual evidence that is consisted of pixel-wise representations of different positions. It may not be suitable to take "visual context" as graph nodes. Because the "visual context" contains complex and uncertain semantics such as different weather and emotional expression, which is different to make sure each node semantics and the number of nodes to represent the whole visual context.

**Q7: Clarification on graph learning, inputs and outpus of the model during training and inference (R3).** The graphs are latent to learn. The inputs of our HGL are candidate answers, a question and an image. The output of the model is a prediction of right answers. The inputs and outputs of the model are the same during training and inference.

**Q8: Explanation for a few questions on section 1 and section 3 (R1 R2 R3).** In section 1, the concept of information isolated island in our paper refers to the independent of different semantic nodes can not achieve semantic inference in a homogeneous graph that connects similar semantic nodes by attribute (e.g. Figure 1(a)) or grammar (e.g. Figure 1(b)). The $y_i^{rl}$ should be $y_i^l$ in Eq.12, we will revise the typos in final version. In L131, the correlations between the vectors are computed via a dot product, we promise to give a more precise definition of this operation in the final version. In section 3.1, maybe "Graph Construction" would be better than the title "Task Definition", we also realize the inappropriate title and will consider carefully the name of the title in our final version. We will briefly describe the three tasks (Q->A, QA->R and Q->AR) and some specific terminologies in our final version to make the paper easier to understand.

**Q9: More qualitative analysis and the application of this work (R3).** Due to the space limit, we will show more qualitative results with analysis in our final version. Our HGL can seamlessly integrate the intra-graph and inter-graph reasoning in order to bridge vision and language domain and can interactively refine reasoning paths for semantic agreement. Such capability would show more potentials in facilitating high-level applications that require cross-domain semantic alignment among visual concepts and linguistic words. (e.g. visual grounding and VQA).

[Meta-Review · NeurIPS 2019]

After considering the author response and discussing this submission, all reviewers recommend acceptance -- including two high ratings. The reviewers generally found the approach novel but were interested in how it applies outside of the VCR task to other question answering datasets. With the addition of the experiments from the rebuttal, this is a strong submission.